# AKT Signaling Modifies the Balance between Cell Proliferation and Migration in Neural Crest Cells from Patients Affected with Bosma Arhinia and Microphthalmia Syndrome

**DOI:** 10.3390/biomedicines9070751

**Published:** 2021-06-29

**Authors:** Camille Laberthonnière, Elva Maria Novoa-del-Toro, Raphaël Chevalier, Natacha Broucqsault, Vanitha Venkoba Rao, Jean Philippe Trani, Karine Nguyen, Shifeng Xue, Bruno Reversade, Jérôme D. Robin, Anais Baudot, Frédérique Magdinier

**Affiliations:** 1Aix-Marseille Univ-INSERM, MMG, 13005 Marseille, France; camille.laberthonniere@univ-amu.fr (C.L.); elva-maria.novoa-del-toro@inrae.fr (E.M.N.-d.-T.); raphael.chevalier@univ-amu.fr (R.C.); natacha.broucqsault@univ-amu.fr (N.B.); jean-philippe.trani@univ-amu.fr (J.P.T.); karine.nguyen@ap-hm.fr (K.N.); Jerome.robin@univ-amu.fr (J.D.R.); anais.baudot@univ-amu.fr (A.B.); 2Department of Biological Sciences, National University of Singapore, Singapore 117558, Singapore; venustas06@gmail.com (V.V.R.); dbsxues@nus.edu.sg (S.X.); 3Département de Génétique Médicale, Hôpital Timone Enfants, 13005 Marseille, France; 4Institute of Molecular and Cell Biology, A*STAR, Singapore 138632, Singapore; bruno@reversade.com; 5Department of Paediatrics, National University of Singapore, Singapore 138632, Singapore; 6Medical Genetics Department, Koç University School of Medicine (KUSOM), Istanbul 34010, Turkey; 7Academic Medical Center (AMC), Reproductive Biology Laboratory, 1012 Amsterdam-Zuidoost, The Netherlands

**Keywords:** SMCHD1, Bosma Arhinia and Microphthalmia Syndrome, Facio Scapulo Humeral Dystrophy, RNA-Seq, neural crest stem cells, induced pluripotent stem cells, systems biology

## Abstract

Over the recent years, the SMCHD1 (Structural Maintenance of Chromosome flexible Hinge Domain Containing 1) chromatin-associated factor has triggered increasing interest after the identification of variants in three rare and unrelated diseases, type 2 Facio Scapulo Humeral Dystrophy (FSHD2), Bosma Arhinia and Microphthalmia Syndrome (BAMS), and the more recently isolated hypogonadotrophic hypogonadism (IHH) combined pituitary hormone deficiency (CPHD) and septo-optic dysplasia (SOD). However, it remains unclear why certain mutations lead to a specific muscle defect in FSHD while other are associated with severe congenital anomalies. To gain further insights into the specificity of SMCHD1 variants and identify pathways associated with the BAMS phenotype and related neural crest defects, we derived induced pluripotent stem cells from patients carrying a mutation in this gene. We differentiated these cells in neural crest stem cells and analyzed their transcriptome by RNA-Seq. Besides classical differential expression analyses, we analyzed our data using MOGAMUN, an algorithm allowing the extraction of active modules by integrating differential expression data with biological networks. We found that in BAMS neural crest cells, all subnetworks that are associated with differentially expressed genes converge toward a predominant role for AKT signaling in the control of the cell proliferation–migration balance. Our findings provide further insights into the distinct mechanism by which defects in neural crest migration might contribute to the craniofacial anomalies in BAMS.

## 1. Introduction

In human cells, the function of the SMCHD1 (structural maintenance of chromosomes flexible hinge domain containing 1) chromatin-associated factor is only partially delineated despite its implication in at least three different rare diseases with distinct clinical features. *SMCHD1* variants were first described in 2–3% of patients affected with Facio Scapulo Humeral Dystrophy [1], a late onset muscular dystrophy with an incidence of 1/20,000 [2] characterized by a predominant involvement of specific muscles of the face, shoulder, and pelvic girdles with progression to peroneal muscles [3]. In FSHD, the majority of patients carry a deletion of an integral number of repetitive macrosatellite elements (D4Z4) in the subtelomeric 4q35 region (type 1 FSHD, FSHD1) [4]. In a smaller proportion of cases, patients carry an *SMCHD1* variant in the absence of D4Z4 array shortening (2–3%; type 2 FSHD, FSHD2) [1].

*SMCHD1* is also mutated in Bosma Arhinia and Microphthalmia Syndrome (BAMS), a rare developmental syndrome characterized by the absence of a nose, olfactory bulbs, microphthalmia, and isolated hypogonadotrophic hypogonadism (IHH) likely associated with a defect in the formation of the nasal placode surrounding neural crest cells migration or impaired projection of the gonadotrophin-releasing hormone (GnRH) neurons during embryogenesis [5,6]. Very recently, a rare *SMCHD1* variant was also identified in a patient with IHH, combined pituitary hormone deficiency (CPHD) and septo-optic dysplasia (SOD), in the absence of arhinia, microphthalmia or muscle involvement [7].

In FSHD, *SMCHD1* missense or splice mutations have been described across the whole coding sequence while in BAMS and IHH-related syndrome, mutations are clustered within exons 3 to 13, spanning a GHKL-type ATPase domain and an associated region immediately C terminal to it [5,6]. Based on functional analysis of the protein domains, it has been proposed that gain-of-function of the ATPase enzymatic activity results in BAMS while loss-of-function or haploinsufficiency is associated with FSHD2 [6,8,9]. However, identical mutations have been shown to segregate with either BAMS or FSHD2 [8,10].

Overall, the wide spectrum of symptoms associated with *SMCHD1* variants points to pleiotropic roles for this protein, depending on cell fate. So far, animal models do not recapitulate the specific features observed in patients, limiting investigations aimed at deciphering circuits involved in the respective diseases. Hence, pathways leading to either FSHD or BAMS and triggering the typical features, i.e., either muscle weakness or defect in neural crest migration, remain undefined [11,12,13,14]. To further characterize disease-specific pathways and understand more specifically neural crest defects linked to *SMCHD1* variants, we derived neural crest cells from induced pluripotent stem cells (hiPSCs) of patients affected with either BAMS or FSHD2 and analyzed their transcriptome by RNA-sequencing. We further analyzed differentially expressed genes with MOGAMUN, an algorithm that uses a multi-objective genetic algorithm to extract active modules by integrating expression data with multiple biological networks. MOGAMUN optimizes two objective functions, one related to interaction density and one related to gene expression deregulation, to reveal active modules, i.e., subnetworks differentially regulated between conditions. We found that mutations found in BAMS cells are associated with changes in the expression of genes encoding extracellular matrix (ECM) components, AKT signaling, neural crest cell migration and cell differentiation, providing insights into the mechanism that might contribute to the craniofacial anomalies in BAMS.

## 2. Materials and Methods

### 2.1. Samples

All individuals provided written informed consent for the collection of samples and subsequent analysis for research purposes. The study was done in accordance with the Declaration of Helsinki. Controls were randomly chosen individuals selected in the same age range and sex representation as patients. Controls neither carried any genetic mutation nor were affected by any constitutive pathology. Samples are listed in Table 1 and described in [9].

### 2.2. Human iPSC Clones

All hiPSC clones were derived from primary fibroblasts (Appendix A) after transfection of different vectors by electroporation (pCXCLE-hOCT3/4-shp53-F (Addgene ref 27077, Watertown, MA, USA), pCXLE-hSK (Addgene, ref 27078, encoding SOX2 and KLF4), pCXLE-hUL (Addgene, ref 27080, encoding L-Myc and LIN28)). Cells were described in [9].

### 2.3. Neural Crest Stem Cells Differentiation

Differentiation was adapted from [15]. Cells were dissociated with Accutase (STEMCELL Technologies, Égrève, France) and plated at a density of 80,000 cells per cm^2^ in hESCs (human Embryonic Stem Cells) medium (DMEMF/12 (Dulbecco’s Modified Eagle Medium), 20% KSR (KnockOut™ Serum Replacement), 1% GlutaMAX, 1% NEAA (Non-Essential Amino Acids), 0.1% β-mercaptoethanol, 20 ng/mL βFGF (Fibroblast Growth Factor)). When cells reached 80% of confluency, differentiation was initiated by addition of KSR medium (KO DMEM, 15% KSR, 1% GlutaMAX, 1% NEAA, 0.1% β-mercaptoethanol) corresponding to day 0 of differentiation (D0). At D0 and D1, KSR medium was supplemented with 500 nM LDN193189 (Cellagen Technologies, San Diego, CA, USA) and 10 µM SB431542 (Merck Chemicals, Darmstadt, Germany). At D2, 3 µM CHIR 99021 (Merck Chemicals, Darmstadt, Germany) was added to the other small molecules. LDN193189 was removed at D3, SB431542 and CHIR 99021 were removed at D4. From D4 to D11, N2 (DMEM/F12, 0.15% Glucose, 1% N2, 20 µg/mL insulin, 5 mM HEPES) was progressively mixed (starting with 15% of N2 at D4) with the KSR medium in order to reach 100% of N2 medium at D10. Cells were collected at D11.

### 2.4. Neural Crest-Like Cells Differentiation

Differentiation protocol was adapted from [16]. iPSC colonies were dissociated into small clusters (50–200 cells) with 0.5 mM EDTA (Sigma Aldrich, Darmstadt, Germany #E6758). The cell clusters were transferred with neural induction media made of 1:1 DMEM/F-12 Glutamax (Gibco #10565-018, Waltham, MA USA) media supplemented with N2 (Gibco #17502048) and Neurobasal media (Gibco #21103-049) supplemented with B27 (Gibco #17504044), 1% Penicillin/Streptomycin, 5 μg/mL Insulin (Sigma Aldrich #I6634), 20 ng/mL bFGF (PeproTech #100-18B, Rocky Hill, NJ, USA) and 20 ng/mL EGF (Sigma Aldrich #E9644) into an ultra-low attachment culture dish (Corning #CLS3261, Corning, NY, USA). Media were changed every day for 12 days till the neuroepithelial rosettes formed. The mature neuroectodermal spheres were then plated on a 1 μg/mL fibronectin (Sigma Aldrich #F1141) coated dish where the spheres attached, and migratory neural crest-like cells emerged.

### 2.5. Immunostaining

NCSCs (Neural Crest Stem Cells) were grown in culture slides until 80% confluency. Cells were fixed using 4% Paraformaldehyde for 20 min at room temperature. After 3 steps of washing with 1× PBS, cells were permeabilized using 0.2% Triton X-100 for 15 min at 37 °C and washed three times with 1× PBS. Non-specific binding was blocked using 0.5% FBS in 1× PBS for 1 h at room temperature. Antibodies against P75 (Millipore 07-476, 1/200, Darmstadt, Germany), SOX10 (R&D Systems MAB2854, 1/200, Minneapolis, MN, USA), Nestin (Chemicon #MAB5326, 1/500, Darmstadt, Germany), and AP2a (Invitrogen #MA1-872, 1/100, Waltham, MA USA) were incubated overnight at 4 °C with gentle shaking. After three washing steps with 1× PBS, slides were incubated with secondary antibodies (Donkey Anti-Rabbit Alexa Fluor^®^ 647, Abcam ab150075, 1/1000, Donkey Anti-Goat Alexa Fluor^®^ 488, Abcam ab150129, 1/1000, Goat anti-Mouse AF 488, Invitrogen #A-11001, 1/500) for 1 h at room temperature. After washing three times with 1× PBS, slides were mounted in VECTASHIELD^®^ Antifade Mounting Medium with DAPI to counter-stain nuclei. Images were acquired using a Widefield fluorescence microscope with optical sectioning Zeiss Apotome.2 (Zeiss, Oberkochen, Germany).

### 2.6. RNA Extraction, Quality Control and Library Preparation

Total RNA was extracted using the RNeasy kit (Qiagen, Düsseldorf, Germany) following manufacturer’s instructions. Quality, quantification, and sizing of total RNA were evaluated using the RNA 6000 Pico assay (Agilent Technologies Ref. 5067-1513, Santa Clara, CA, USA) on an Agilent 2100 Bioanalyzer system. The RNA integrity number (RIN) was calculated for each sample and only samples with an RIN >9 were kept for further use. Libraries were constructed using 2 µg of total RNA. The TruSeq Stranded mRNA Library Preparation Kit High Throughput (Illumina, ref RS-122-2103, San Diego, CA, USA) was used according to the manufacturer’s guidelines. Briefly, PolyA + containing RNA molecules were purified using polyT oligo-attached magnetic beads. Thermal fragmentation was carried out after two rounds of enrichment for PolyA + mRNA. cDNA was synthesized using reverse transcriptase (Superscript II) and random primers. This was followed by second strand cDNA synthesis, end repair process, adenylation of 3′ ends, and ligation of the adapters. The products were then purified and enriched with 15 cycles of PCR to create the cDNA library. Libraries were quantified by qPCR using the KAPA Library Quantification Kit for Illumina Libraries (Roche, ref. 7960140001). Library profiles were assessed using the DNA High Sensitivity LabChip Kit (Agilent Technologies Ref. 5067-4626) on an Agilent Bioanalyzer 2100. Libraries were sequenced on an Illumina NextSeq 500 System using a cartridge of the NextSeq 500/550 High Output v2 kit (150 cycles) (Illumina FC-404-2002).

### 2.7. RNA-Seq Data Processing and Differential Expression Analysis

We assessed FASTQ sequence data quality using FastQC v0.11.5 and trimmed the reads to remove adapter sequences and low-quality bases using Trimmomatic v0.36 [17]. The resulting trimmed paired-end reads were aligned using STAR v2.5.3a [18] to the GRCh38 human genome release. Obtained BAM files were indexed using Sambamba (v0.6.6) after ordering them by coordinates. Aligned reads were counted with StringTie v1.3.1c using GENCODE annotation. Differentially expressed genes (DEGs) of different conditions were identified using the DESeq2 (v1.18.1) R-package.

The number of raw sequence reads, remaining reads after trimming, and mapping rates for each sample are shown in Appendix A.

Differentially expressed genes were extracted at the fold-change cutoff ≥2 and an FDR (False Decovery Rate)-corrected *p*-value of 0.05 (named after FDR). We used the Ensembl human gene IDs identified in the DESeq2 analyses as input for further analyses.

The raw RNA-Seq data and raw count matrix were deposited at the NCBI Gene Expression Omnibus (https://www-ncbi-nlm-nih-gov.gate2.inist.fr/geo/ (accessed on 23 June 2021) under the accession GSE173251.

### 2.8. Gene Ontology Analysis

Overrepresentation test analyses were performed using enrichGO from the R-package clusterProfiler (v3.10.15). We identified biological pathways (BPs) with an FDR < 0.05. Results are presented as bar plots, where the *x*-axis represents the log10(FDR), the corresponding GO terms are shown on the *y*-axis, and the percentage of DEGs associated with a given GO term is at the top of each bar plot.

### 2.9. Quantitative RT-PCR

Reverse transcription of 1 µg of total RNA was performed using the Superscript IV First-Strand cDNA Synthesis kit (Life Technologies, Waltham, MA USA). We used a mix of oligo dT and random hexamers to target all types of RNA and followed manufacturer’s instructions to synthesize cDNA. The remaining RNA was degraded by incubation with *Escherichia coli* RNase H (Life Technologies) at 37 °C for 20 min. Primers were designed using Primer Blast and Primer3 (Appendix A). Real-time PCR amplification was performed on a LightCycler 480 (Roche) using the SYBR green master mix. All PCRs were performed using a standardized protocol and data were analyzed with the LightCycler 480 software version 1.5.0.39 (Roche). Primer efficiency was determined by absolute quantification using a standard curve. For each sample, fold change was obtained by comparative quantification and normalization to expression of the *HPRT*, *GAPDH* and *PPIA* housekeeping genes used as standard. Data are expressed as means ± SD.

### 2.10. RNA-Seq Data Analysis

RNA-Seq data were analyzed using MOGAMUN, a multi-objective genetic algorithm that identifies active modules (i.e., highly interconnected subnetworks with an overall deregulation) in a multiplex biological network (i.e., a network composed of different layers, where each layer represents physical and/or functional interactions).

We performed an analysis of differential expression of our RNA-Seq data, to compare patients vs. controls with DESeq2. We used as input for MOGAMUN the resulting FDR-corrected *p*-values, and a multiplex biological network composed of three layers or undirected interactions from [19]. The first layer corresponds to physical protein–protein interactions, and it was obtained by merging the CCSB Interactome database [20] and several databases from the PSICQUIC portal [21]. The second layer contains data of pathways, obtained using the R-package graphite [22]. The third layer corresponds to the absolute Spearman correlation superior or equal to 70%, calculated using the RNA-Seq expression data of 32 tissues and 45 different cell lines [23]. We ran MOGAMUN 30 times with the default parameters. The resulting accumulated Pareto front was composed of several active modules that contained different numbers of genes, including significant ones (considering an absolute log fold change >1 and FDR <0.05). Active modules were visualized using Cytoscape v.3.8.2.

Genes corresponding to each node were analyzed using g: Profiler (https://biit.cs.ut.ee/gprofiler (accessed on 23 May 2021)) [24] to define the corresponding molecular function and obtain the corresponding *p*-value. g: Profiler takes as input a list of genes, uses their annotation (from the Ensembl database), and performs a hypergeometric test to find over-representation (i.e., enrichment) of functional terms (e.g., biological pathways, molecular functions, cellular components, etc.). Here we used g: GOSt to perform the functional enrichment analysis of individual gene lists.

### 2.11. Wound Healing Closure Scratch Test

NCSCs (Neural Crest Stem Cells) were seeded in 12-well plates coated with Matrigel. At 90–100% confluency, the scratch was made using a pipette tip. Live imaging was performed using a Fast-Imaging Observer (Zeiss, Oberkochen, Germany) with an incubation chamber set at 37 °C and 5% CO_2_. Images were taken every 15 min during 25 h.

Analysis of gap areas was performed using the ImageJ software and the MRI (Montpellier Ressources Imagerie) wound healing tool plugin. The last 5 h were excluded since the image analysis fails to detect and measure gap area when gaps were nearly closed. For each condition, mean gap area was plotted as a function of time and compared using a Friedman statistical test.

### 2.12. Flow Cytometry Analysis

Cells were collected with Accutase, resuspended in 1X PBS + 5% FBS (Fetal bovine serum) and incubated with APC-conjugated (Allophycocyanin) anti-PDGFRα antibody (Platelet-Derived Growth Factor Receptor Alpha, 1/100, 130-115-239, Miltenyi Biotec, Bergisch Gladbach, Germany) or PE-conjugated (Phycoerythrin) anti-PDGFRβ antibody (Platelet-Derived Growth Factor Receptor Beta, 1/100, 130-123-772, Miltenyi Biotec, Bergisch Gladbach, Germany). After 30 min of incubation, cells were washed three times by centrifugation and resuspended in 1× PBS + 5% SVF. The analysis was performed with an ACCURI C6 flow cytometer on the basis of Forward Scatter (FSC-A) versus side scatter (SSC-A) for the selection of living cells and the elimination of cell aggregates or debris. Analysis of fluorescent population was limited to living cells. Unstained cells and isotype controls were used to determine the background of fluorescence and compensation was determined for individual fluorochromes.

## 3. Results

### 3.1. Experimental Design

In order to identify genes that are dysregulated in neural crest cells, we differentiated hiPSCs from patients with FSHD (type 1 or 2) or BAMS [9] toward neural crest stem cells (NCSCs) [15] and compared gene expression to healthy donors at day 11 post-differentiation (Figure 1A). For patients carrying an *SMCHD1* variant, we compared cells from a patient affected with BAMS carrying a mutation (BAMS-1; E136G) that causes an increased ATPase enzymatic activity, reported as a gain-of-function [6,8] to cells from a patient affected with FSHD2 carrying a mutation in the ATPase domain, never reported in BAMS and which abrogates SMCHD1 enzymatic activity (14586; FSHD2, Q194P) [9] (Figure 1B). To identify pathways that distinguish BAMS from FSHD cells, we also included in our analyses cells from patients affected with type 1 FSHD (FSHD1), i.e., one patient carrying a short D4Z4 allele (#12759; 7 Repetitive Units, (RUs)) and one patient with mosaicism (#17706; 25% of mosaicism, clinically affected) for whom a clone with a short allele (2 RU, 4qA allele) and a clone with a long allele (healthy allele, 15 RU; other 4qA allele: 41 RU) were selected [9]. For each sample, RNA-sequencing was performed on two biological replicates corresponding to two independent differentiation experiments at day 11 of differentiation. At this stage, the vast majority of cells expressed the Neurotrophin Receptor P75 (*p75^NTR^*) neural crest cell marker and SOX10, a marker of late pre-migratory neural crest cells (Figure 1C), together with other neural crest markers (Appendix A) indicating a high differentiation efficiency.

In patients’ cells, Differentially Expressed Genes (DEGs) were selected based on a minimum 2-fold change and statistical significance of *p* < 0.05 compared to control NCSCs. We retrieved 599 DEGs only found in FSHD1 NCSCs, 684 in FSHD1 mosaic cells (compared to its isogenic control), 576 in FSHD2, and 657 in BAMS NCSCs (Figure 1D, Appendix A). FSHD1 and FSHD2 shared 142 unique DEGs. FSHD1 and FSHD1 mosaic shared 196 unique DEGs, 274 were common between FSHD1 and BAMS cells and 505 between BAMS and FSHD2 (Figure 1D, Appendix A) with a majority of upregulated genes (Appendix A) and a high proportion of long non-coding RNAs (LncRNAs) in FSHD1 and FSHD1 mosaic NCSCs (Appendix A).

Gene Ontology (GO) analysis for FSHD1 (Figure 1E), FSHD1 mosaic (Figure 1F), and FSHD2 DEGs (Figure 1G) revealed enrichment in biological pathways (BPs) corresponding to muscle development or functions such as “muscle tissue morphogenesis”, “muscle organ development”, “muscle contraction”, “muscle structure development” or “muscle system process”. Interestingly, consistent with hearing loss in FSHD patients [25] and the implication of a number of muscle genes in decreased sound perception, the BP term corresponding to “ear development” was also represented. By looking more closely at DEGs shared between FSHD1, FSHD1 mosaic, and FSHD2 NCSCs (Figure 1H), we found five cellular components (CCs) common to the three sample categories. Among them, four were related to the extracellular compartment. Consistent with the neural crest origin of melanocytic cells, the fifth cellular component was related to melanosomes membrane lipid bilayer and might be overrepresented here due to three DEGs (*DCT*, *GPR143,* and *TH*) over the 12 genes corresponding to this cellular compartment (Figure 1H).

### 3.2. DUX4 and DUX4-Target Genes Are Detectable in FSHD and BAMS Neural Crest Cells

In FSHD, hypomethylation of D4Z4 and subsequent chromatin relaxation has been associated with ectopic expression of *DUX4* encoded by D4Z4, considered as causative of FSHD [26,27]. We previously showed that the pathogenic *DUX4-fl* transcript as well as some of the 422 target genes activated by this transcription factor, such as *ZSCAN4*, *TRIM43* or *MBD3L2,* are produced at a comparable level in FSHD1, FSHD2, and BAMS pluripotent cells [9]. We first tested whether DUX4 and its target genes [28] were among the NCSC DEGs. Given its low expression level, *DUX4* transcripts are not detectable by RNA-Seq in NCSCs, as reported by others in muscle cells [29,30], but they are detectable by RT-qPCR in FSHD2 (*p*-value < 0.0005), FSHD1 mosaic (*p*-value < 0.05), and at a low level in FSHD1 and BAMS NCSCs (Figure 2A). By comparing the list of DUX4 target genes [28] to FSHD1, FSHD2, and BAMS DEGs in NCSCs, we found that the vast majority (351/422 genes) are not differentially expressed in patients’ NCSCs. Only 13 DUX4 targets are deregulated in the three diseases (Figure 2B) with eight genes upregulated (2.2–5.6 fold-change range) and five genes downregulated (2.5–7.65 fold-change range) in the different conditions (Figure 2C). We then verified expression changes for genes considered as consistent markers of DUX4 activation by RT-qPCR (Figure 2D). We only detected a significant difference in expression for *ZSCAN4* in FSHD1 mosaic compared to its isogenic control NCSCs (*p*-value < 0.05) and a significant difference for *LEUTX* between FSHD1 mosaic or FSHD1 cells compared to controls (*p*-value < 0.005). Compared to controls, the expression of the other genes (*ZSCAN4*, *MBD3L2*, *TRIM43*) was highly variable between samples (Figure 2D). Thus, as reported before [9], *DUX4* and a number of its target genes are expressed in BAMS and FSHD non-muscle cells indicating that induction of this retrogene and associated targets is not muscle-specific and that BAMS patients are permissive for *DUX4* transcription despite the absence of muscle defects that are specific to the FSHD phenotype [6].

### 3.3. BAMS-Specific DEGs Are Associated with Cell Migration and Communication

As NCSCs contribute to the development of facial structures, which are strongly impacted in BAMS patients, we looked more specifically at the 657 DEGs that are unique to BAMS cells (Figure 3A). In BAMS NCSCs, BPs associated with these DEGs correspond to, among others, “cell development”, “cell morphogenesis”, “axon guidance”, “telencephalon cell migration” and “neural crest cell migration” pathways (Figure 3A). BPs corresponding to “embryonic eye development” and “cartilage development” also appear highly interesting in light of BAMS patients’ phenotypes. BPs related to “pattern specification process” and “extracellular matrix organization” are shared between FSHD and BAMS cells.

Consistent with the impaired axonal projection, cellular component enrichment analyses retrieved “post-synapse”, “synapse”, “neuron projection”, and “neural crest cell migration” GO terms (Figure 3B). Among the DEGs related to “neural crest development” and also “neuron projection” GO terms, seven semaphorins were dysregulated, with three of them upregulated in BAMS and FSHD2 NCSCs (*SEMA3A*, *SEMA3D*, and *SEMA3E*) and four downregulated in FSHD2 and BAMS NCSCs (*SEMA4C*, *SEMA5B*, *SEMA6B*, and *SEMA7A*). Semaphorins are extracellular signaling proteins that are essential for development and tissue maintenance and play key roles in the development of many organs [31,32], axonal guidance, and craniofacial development [33,34]. In addition to semaphorins, we observed changes in several factors that belong to the three other classes of canonical axon guidance molecules and control of axonal growth, Netrins, Ephrin, Slits and Slit receptors [35,36] (Appendix A), consistent in BAMS patients with a possible defect in the projection of the gonadotrophin-releasing hormone neurons during embryogenesis [6].

### 3.4. Neural Crest Stem Cell (NCSC) Differentiation Reveals Impairment of Extracellular Matrix Synthesis

Complementary to classical RNA-Seq analyses, the expression data were analyzed with MOGAMUN [15], using the lists of DEGs as seeds. MOGAMUN is a recently developed algorithm that integrates expression data with multiplex biological networks to identify active modules. For our experiments, we considered protein–protein interactions (blue edges), biological pathways (red edges), and correlation of expression (yellow edges) in the multiplex biological networks. Depending on the samples, 9 to 19 subnetworks (active modules) were retrieved. We first selected similar gene sets, studied their information content, and grouped the subnetworks based on their associated BP terms, determined using g:Profiler.

Together with BP terms highlighting the implication of cell membrane components associated with “cell junction”, “cell periphery”, “intrinsic component of membrane”, and “membrane” in BAMS, FSHD1 and FSHD2 NCSCs, network analysis retrieved several pathways linked to extracellular matrix (ECM) organization and cell adhesion (Figure 3C–H, Appendix A). In FSHD1 cells in particular, 14 out of the 17 active modules indicated an upregulation of several members of the collagen, laminin, or integrin families. On the other hand, BAMS cells displayed a global decrease in expression of genes encoding these same ECM components (Figure 4E–H; Appendix A), suggesting a “mirror” effect in the composition or stiffness of the ECM between the two diseases.

### 3.5. Altered PI3K/AKT Signaling is Specific to BAMS Cells

To identify biological processes specific to each condition, we then searched for genes that are consistently enriched in other active modules and associated BPs. Top-perturbed pathways in BAMS NCSCs revealed alteration in PI3K/AKT signaling. Furthermore, BAMS subnetwork analysis converged toward a defective cell proliferation with a strong upregulation of the *PIK3R2* oncogene (Log FC 11.11; *p*-value = 1.77 × 10^−12^) and downregulation of *PIK3R1* (Log FC −1.37; *p*-value 0.011), inducing matrix degradation and cell proliferation *(*Figure 4A–D, Appendix A). In addition and as observed for ECM components, we observed opposite expression profiles between BAMS and FSHD2 cells for *MET* required for cell migration [37,38], downregulated in BAMS (Figure 4A–D, Log FC −7.6; *p*-value = 4.72 × 10^−66^) but upregulated in FSHD2 (Figure 4E–H, Log FC 5.9; *p*-value = 1.5 × 10^−17^) or *ERBB3* required for cell–cell adhesion and motility (Figure 4B–D, Log FC −2.9; *p*-value = 2.6 × 10^−20^ in BAMS; Log FC 5.52, *p*-value = 2.13 × 10^−91^ in FSHD2, Figure 4E–G).

Further consistent with the involvement of PI3K/AKT signaling, Platelet-Derived Growth Factor Receptors (PDGFRs) α and β genes that bind different growth factors at the cell surface were differentially expressed in the two cell types with a decreased expression of *PDGFRA* and *B* (Figure 4B–D, Appendix A) in BAMS (Log FC −2.5, *p*-value = 4.24 × 10^−21^; Log FC −3.2, *p*-value = 6.7× 10^−86^, respectively) but an upregulation in FSHD2 cells (Log FC 3, *p*-value = 1.84 × 10^−33^; Log FC 3.22, *p*-value = 5.78× 10^−51^, respectively, Figure 4G–H, Appendix A) and FSHD1 (Appendix A).

In light of the BAMS phenotype, we also identified another interesting factor, *NEDD4* encoding the neural precursor cell expressed, developmentally downregulated E3 Ubiquitin Protein Ligase and required for cranial neural crest stem cell survival and dynamics [39]. This gene was downregulated in BAMS (Figure 4D, Log FC −2.51, *p*-value = 5.32 × 10^−14^) but increased in FSHD2 (Figure 4G, Log FC 2.52, *p*-value = 9 × 10^−17^) or FSHD1 (Appendix A, Log FC 1.80, *p*-value = 8.27 × 10^−8^).

Additional subnetworks were related to desmosome organization (FSHD1, Appendix A), positive regulation of signal transduction (Figure 4G, Appendix A). Depending on the sample, two to nine active modules also highlighted the downregulation of proteins involved in DNA replication, which might be the results of the proliferative states of the cells (Appendix A).

### 3.6. BAMS Neural Crest Cells Display a Defect in Cell Migration

Given the redundancy of PDGFRα and β in the different active modules retrieved by MOGAMUN and their implication in neural crest cell migration, we confirmed their marked decreased expression by RT-qPCR in NCSCs derived from additional BAMS patients (Figure 5A). By analyzing the proportion of NCSCs expressing PDGFRα, β or both in control and BAMS cells from three different patients by flow cytometry and immunostaining at day 11 post-differentiation, we observed a significant decrease in the proportion of PDGFRβ-positive cells in BAMS cells but a similar percentage of PDGFRα-positive cells. We further showed that the proportion of cells expressing both membrane receptors was also significantly decreased in BAMS cells (0.2 to 14.5%) compared to controls (37.9%). Confirming RNA-Seq data we also observed a marked decrease in the proportion of cells expressing ERBB3 between controls (75.9%) and BAMS cells (1.8–7.2%) (Figure 5B).

Altogether, downregulation of *PDGFRα* or *β* and *ERBB3*, *cMET* and integrins or changes in ECM composition converged toward a defect in cell–cell contacts, cell-substratum adhesion, and cell migration in BAMS NCSCs. To address this hypothesis, we first quantified the proportion of proliferative Ki67-positive cells by flow cytometry and did not evidence any significant difference between conditions (Figure 5C). We then analyzed NCSC migration capacity by performing scratch tests closure and recording of live cell migration and wound healing (Figure 5E). Scratches were made on confluent monolayer NCSCs using a pipette tip, and live cells were scanned for 20 h at different time intervals (Figure 5E). Comparison of the closure area at each time point revealed significant differences between conditions with a slower scratch closure in BAMS cells compared to controls (Figure 5F). We further corroborated these findings using an alternative differentiation protocol for neural crest stem cells based on a neurosphere differentiation step (Figure 5G) [16] by showing a decrease in the overall migration of BAMS-derived cells (Figure 5H).

Altogether, these data confirmed the possible defect in cell migration in BAMS cells and its possible link with PI3K/AKT signaling and decreased expression of membrane receptors responsible for cell–cell and cell-to-matrix interactions.

## 4. Discussion

One of the most striking phenotypical features in BAMS patients is the arhinia and olfactory bulbs absence [6]. During development, the olfactory bulbs arise from the olfactory placode and the multipotent neural crest cells [40] that migrate throughout the embryo and differentiate into a broad range of cell types including cranial cartilages and bones [41]. BAMS congenital malformations are suspected to arise from a defective migration of cells emanating from the nasal placode or impaired projection of GnRH neurons [5,6]. However, key genes and pathways linking *SMCHD1* mutation to this congenital anomaly in BAMS remains elusive. In order to identify pathways leading to the BAMS phenotype and associated with *SMCHD1* variants, we differentiated control, BAMS hiPSCs into NCSCs together with cells from patients affected with type 1 or 2 FSHD [15] and performed transcriptome analysis at day 11 of differentiation.

As network-based approaches have been highly efficient for identification of biological processes perturbed in diseases, we performed in-depth analysis of our RNA-Seq data using MOGAMUN, a novel algorithm designed to identify active modules by integrating expression data with one or more networks of biological interactions [42]. By comparing the different active modules found in the different conditions using DEGs as seeds, we identified genes and pathways that appeared highly meaningful with regard to neural crest cell differentiation and BAMS pathophysiology. Indeed, MOGAMUN retrieved several networks highlighting changes in the extracellular matrix organization and related function in cell adhesion in the different conditions. Markedly in BAMS, the top-perturbed pathways implicated PI3K/AKT, PDGFR, and ERBB3 signaling, with a striking mirror effect between BAMS (downregulation) and FSHD2 or FSHD1 (upregulation) cells.

Cell migration is a complex process that requires a coordinated response to extracellular stimuli and cellular activities. Using RNA-Seq, we showed that BAMS NCSCs are characterized by an overall decrease in the expression of a number genes required for NCSC migration and confirmed this finding in live cells.

PI3K/AKT signaling is involved in at least three types of cellular modifications during neural crest cell differentiation, including cytoskeleton changes, stability of cell–cell junctions, and cell-to-substratum interactions [43], all three represented in subnetworks retrieved in BAMS cells. Consistent with a role for AKT signaling in velocity, directionality, and distance of migrations [44,45] and possible defects in NCSC migration in BAMS patients, we observed an upregulation of *PIK3R2* involved in cell proliferation but matrix degradation [46] and downregulation of *PIK3R1* that regulates cell migration. In the mouse, alterations in PI3K/AKT signaling are associated with the absence of PDGFRs [47,48], also downregulated in BAMS NCSCs. PDGFRα and β receptor Tyrosine kinases are involved in a broad range of cellular and developmental processes such as cranial development [47,49,50]. Upon activation, PDGFRs form either homo- or α/β heterodimers leading to phosphorylation of Tyrosine residues at specific sites [51] and activation of different substrates such as Phosphatidyl Inositol 3 Kinase. The PDGFR-activated PI3K/AKT pathway promotes Actin recombination, stimulates cell growth, and inhibits cell apoptosis. The neural crest lineage invalidation of PDGFRα results in facial anomalies [52] and delayed migration into the frontonasal prominence [53]. PDGFRβ is also expressed during embryogenesis and required for nasal development associated with dysregulation or mislocalization of various ECM proteins [47,54,55]. PDGFRα and β functions are not overlapping in the developing frontonasal region in the mouse. During development, PI3K is the main downstream effector of PDGFRα in this species. Interestingly, and as observed in PDGFRα or β conditional knock-out mouse models [47], we observed that changes in *PDGFR* expression correlate with modifications in the expression of proteins involved in ECM organization, including many collagens or integrins required for cell–ECM adhesion, with opposite gene expression profiles in BAMS and FSHD2 cells. Consistent with our hypothesis of the possible involvement of PDGFR and PI3K/AKT signaling in facial anomalies in BAMS, mutations in PDGFRs are associated with syndromes characterized by facial dysmorphism [56,57]. Likewise, in BAMS, we observed a slight but non-significant decrease in *PTPN11* expression (FC −0.69, *p*-value = 6.01 × 10^−06^), encoding SHP-2 that binds to PDGFRα and is also associated with facial dysmorphism in Noonan syndrome [58], further underlining the importance of this signaling pathway in BAMS.

Overall, our results in NCSCs derived from patients affected with BAMS suggested that during development, variable *PDGFR* expression and the proportion of NCSCs expressing α and β PDGFRs might be involved in AKT-dependent signal transduction capacities, ECM composition, and cell-to-matrix contacts, thereby influencing cell migration. Studies in various animal models showed that defective AKT signaling affects neural crest cell (NCC) migration and results in a limited population of NCCs reaching their target tissue. Our results also revealed changes in c-MET expression in BAMS cells, another key gene involved in neural crest migration in response to HGF stimulus [59] or *NEDD4*, required to maintain survival of cranial neural crest cells [39].

Our data indicated that cell proliferation and survival is not altered in BAMS cells while cell migration might be at least partly reduced through downregulation of a number of key genes such as cMET, PDGFRs, and NEDD4, among others. Thus, based on the pathways uncovered by a system biology approach and in vitro validation, our results provide a potential explanation for the facial alterations and impaired projection of the gonadotrophin-releasing hormone neurons occurring in BAMS that depend on changes in the balance of genes controlling proliferation and genes controlling migration and cell motility. Furthermore, defects in PAX3-positive neural crest and defects in ECM stiffness and composition might also explain alterations in subcutaneous facial and trunk muscles in FSHD patients.

Of note, we analyzed expression profile in two types of cells associated with disease phenotype in FSHD (muscle fibers, Laberthonnière et al. submitted) and BAMS (neural crest cells). Since during development neural crest cells and skeletal muscle lineage are established in a synchronous manner [60], it is interesting to note that different DEGs are associated with pathways that are common to conditions with mirror expression levels between the two diseases, thus providing instructive insights into their respective pathomechanisms.

## Figures and Tables

**Figure 1 biomedicines-09-00751-f001:**
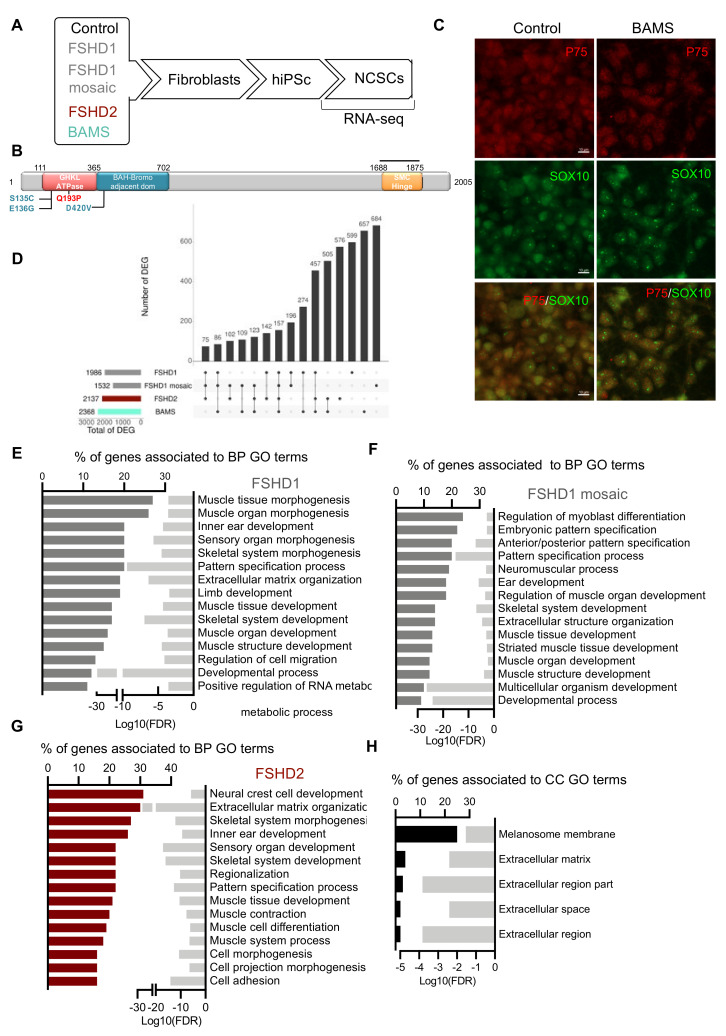
Gene expression profiling in Facio-Scapulo-Humeral Dystrophy (FSHD) patients’ neural crest stem cells. (**A**) Schematic overview of the samples used and steps of analysis. Induced pluripotent stem cells (hiPSCs) were derived from primary fibroblasts from healthy donors (controls), patients affected with type 1 FSHD (one FSHD1 patient carrying 7 D4Z4 repeated units (#12759) and one patient with mosaicism (25% of the cells carry a 4qA allele with 2 D4Z4 repeated units; 75% of cells carry a 4qA allele with 15 D4Z4 repeated units (#17706)), FSHD2 (#14586; c.573A > C; p.Q193P) or Bosma Arrhinia and Microphthalmia Syndrome (BAMS) (BAMS-1; c.407A > G; p.E136G) (Appendix A). Fibroblasts were reprogrammed into human induced pluripotent stem cells (hiPSCs). For FSHD1 mosaic cells, hiPSC clones carrying the contracted D4Z4 or the isogenic non-pathogenic D4Z4 allele were isolated separately. Cells were described in [9]. All hiPSC clones were differentiated into Neural Crest Stem Cells (NCSCs) as described [15]. Gene expression analysis was performed by high throughput RNA-sequencing at day 10 of differentiation; (**B**) Schematic representation of the SMCHD1 (Structural Maintenance of Chromosome flexible Hinge Domain containing 1) protein and position of mutations in BAMS (cyan) or FSHD2 patient (red). BAMS-1 carries a missense mutation in the ATPase domain reported as a gain-of-function (E136G). FSHD2 patient #14586 carries a mutation in the ATPase domain reported as a loss-of-function of the ATPase activity (Q193P); (**C**) Immunostaining of NCSCs stained with antibodies against p75NTR (green) and SOX10 (red) at day 10 post-differentiation; (**D**) Upset plot for comparison of Differentially Expressed Genes (DEGs) with fold change >2 or <−2 and a *p*-value < 0.05 in FSHD1, FSHD1 mosaic, FSHD2 and BAMS NCSCs; (**E**–**H**) Gene Ontology (GO) terms for biological pathways (BPs) corresponding to enrichment analysis of unique DEGs. The bars on the left represent the percentage of DEGs determined for each condition from the total number of DEGs. Light gray bars on the right represent the enrichment score (Log10 of False Discovery Rate) for each GO term. (**E**) FSHD1 vs. control NCSCs (dark gray bars); (**F**) FSHD1 mosaic (dark gray bars) vs. isogenic control NCSCs; (**G**) FSHD2 vs. control NCSCs. (dark red bar plot); (**H**) GO terms associated with Cellular Component (CC) analysis for DEGs common to FSHD1, FSHD1 mosaic, and FSHD2 NCSCs.

**Figure 2 biomedicines-09-00751-f002:**
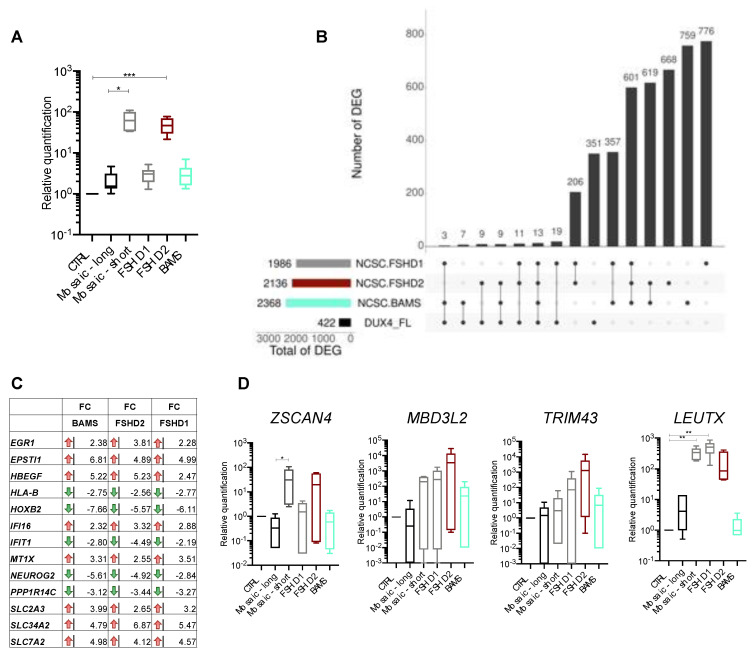
FSHD and BAMS neural crest cells express DUX4 and DUX4 target genes. (**A**) RT-qPCR for *DUX4* expression in NCSCs derived from the different samples. Expression was normalized to the expression level of three different housekeeping genes (*GAPDH*, *HPRT* and *PPIA)* and to the expression in controls (ΔΔCT method, n = 6; technical triplicates in biological duplicates). Means ± SD and statistical significance are reported; Kruskal–Wallis multiple comparison test, * *p*-value < 0.05, *** *p*-value < 0.0005); (**B**) Upset plot for comparison of DEGs in FSHD1 (gray), FSHD2 (orange), BAMS (cyan) to the list of 422 genes regulated by *DUX4* overexpression [28] in NCSCs. We retrieved 19 DUX4-fl target genes specifically deregulated in FSHD1 NCSCs, 9 in FSHD2 NCSCs and 7 in BAMS NCSCs. Thirteen DEGs are common to FSHD1, FSHD2 and BAMS; (**C**) List of DUX4 target genes that are differentially expressed in the different pathologies with red arrows corresponding to genes that are downregulated and green arrows, to upregulated genes. The fold change is indicated for all of them. DUX4 target genes dysregulated in all three conditions (FSHD1, FSHD2, and BAMS) with 8 genes significantly upregulated (green arrows, *EGR1*, *EPSTI1*, *HBEGF*, *IFI16*, *MTX1*, *SLC2A3*, *SLC34A2*, *SLC7A2*) and 5 genes downregulated (*HLA-B*, *HOXB2*, *IFIT1*, *NEUROG2*, *PPP1R14C*); (**D**) Validation by RT-qPCR of selected DUX4 target genes expression in NCSCs. Box plots display the results of biological and technical triplicates for each group of samples (Controls, FSHD1, FSHD1 short corresponds to the clone containing the contracted D4Z4 allele for the mosaic patient and FSHD1 long corresponds to its isogenic control; FSHD2 and BAMS). Statistical significance was determined using a Kruskal–Wallis statistical test. * *p*-value < 0.05, ** *p*-value < 0.005.

**Figure 3 biomedicines-09-00751-f003:**
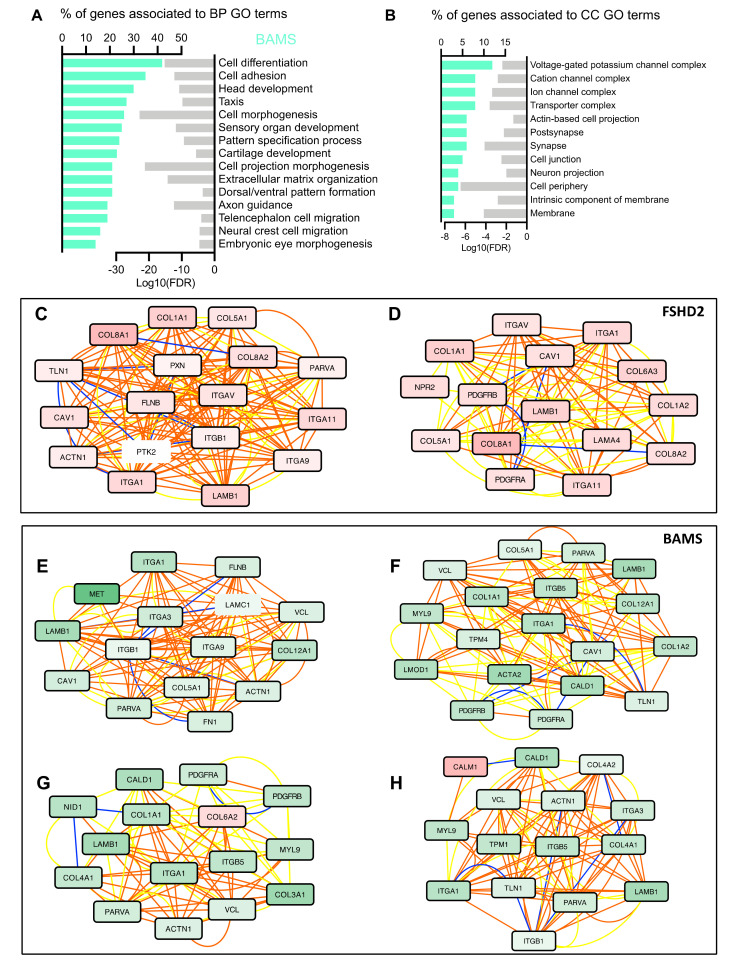
Differential expression analysis in BAMS NCSCs reveals defects in cell adhesion and extracellular matrix organization. (**A**) BP corresponding to enrichment analysis of unique DEGs (657 genes) in BAMS vs. control NCSCs filtered on 2 > FC > 2 and *p*-value < 0.05. Bar plot on the left represents the percentage of genes that are deregulated and associated with a BP GO-term (Gene Ontology) shown in the right column. Light gray bars on the right represent the enrichment score (Log10 of False Discovery Rate) for each GO term. (Dark red bar plot); (**B**) GO terms associated with Cellular Component (CC) analysis for DEGs common to FSHD1, FSHD1 mosaic, and FSHD2 NCSCs; (**C**–**H**) Representative active modules sampled from the accumulated Pareto front of 30 runs for the different data sets using the MOGAMUN algorithm. The color of the edges (links) denotes the type of interaction/relationship between each pair of genes, specifically, protein–protein interactions (blue links), biological pathways (orange links), and correlation of expression data (yellow links). Upregulated nodes are colored in red and downregulated ones in green, the intensity of the color reflects the fold change. Nodes with a dark border are significant (i.e., False Decovery Rate (FDR) < 0.05 and −2 > FC > 2). (**C**,**D**) Active modules containing genes related to Extracellular Matrix (ECM) organization and function in FSHD2 NCSCs with a majority of upregulated genes. (**E**–**H**) In BAMS NCSCs, MOGAMUN uncovered different active modules comprising genes related to ECM organization and functions, with a majority of downregulated genes, except for *COL6A2* (**E**) or *CALM1* (**F**). (**C**–**G**) The list of genes in each node was analyzed using g:Profiler to define the corresponding molecular function and *p*-value. All nodes correspond to ECM organization (**C**) *p*-value = 9.98 × 10^−11^. (**D**) *p*-value = 3.9 × 10^−11^. (**E**) *p*-value = 7.06 × 10^−14^. (**F**) *p*-value = 3.3 × 10^−8^. (**G**) *p*-value = 1 × 10^−13^. (**H**) *p*-value = 3.6 × 10^−17^.

**Figure 4 biomedicines-09-00751-f004:**
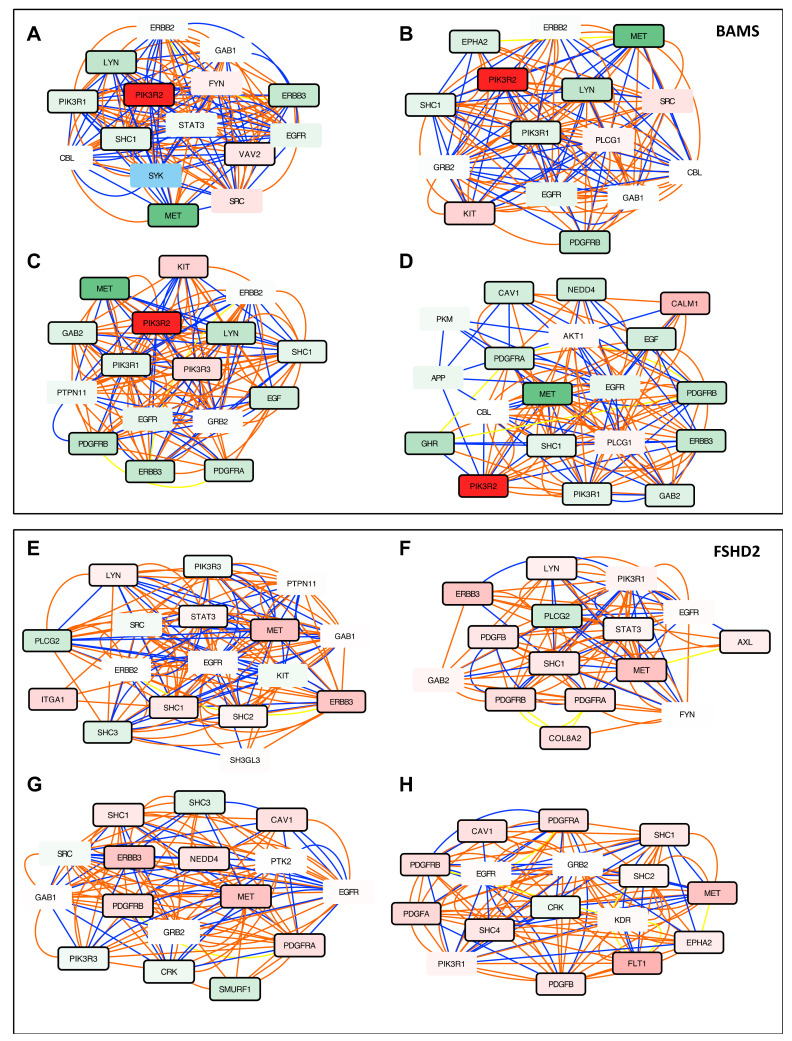
Networks analysis of SMCHD1-mutated cells reveals defects in cell signaling. Representative active modules sampled from the accumulated Pareto front of 30 runs for the different data sets using the MOGAMUN algorithm. The color of the edges (links) denotes the type of interaction/relationship between each pair of genes, specifically, protein–protein interactions (blue links), biological pathways (orange links), and correlation of expression data (yellow links). Upregulated nodes are colored in red and downregulated ones in green, the intensity of the color reflects the fold change. The black border reflects the level of significance (FDR < 0.05 and −2 > FC > 2). Genes corresponding to each node were analyzed using g:Profiler to define the corresponding molecular function and *p*-value. A large majority of nodes in BAMS and FSHD2 cells highlight changes in transmembrane receptor protein Tyrosine kinase signaling. Genes related to Tyrosine Kinase signaling: *PIK3R1*, *PIK3R2*, *ERBB2*, *ERBB3*, *MET*, *EGF*, *EGFR*, *SHC1*, *SRC*, *VAV2*, *SYK*, *GAB1*, *CBL*, *LYN*, *EPHA2*, *CALM1*, *PLGC1*, *APP*, *AKL*, *PTPN11*, *SHC2*, *EPHA2*, *PLT1.* (**A**–**D**) Representative nodes corresponding to Tyrosine Kinase signaling identified in BAMS NCSCs. (**A**) *p*-value = 3.3 × 10^−18^. (**B**) *p*-value = 3.3 × 10^−18^. (**C**) *p*-value = 1.3 × 10^−19^. (**D**) *p*-value = 1.3 × 10^−7^. (**E**–**H**) Representative nodes corresponding to Tyrosine Kinase signaling in FSHD2 NCSCs. (**E**) *p*-value = 1.14 × 10^−17^. (**F**) *p*-value = 2.4 × 10^−18^. (**G**) *p*-value = 8.5 × 10^−15^. (**H**) *p*-value = 1.14 × 10^−15^.

**Figure 5 biomedicines-09-00751-f005:**
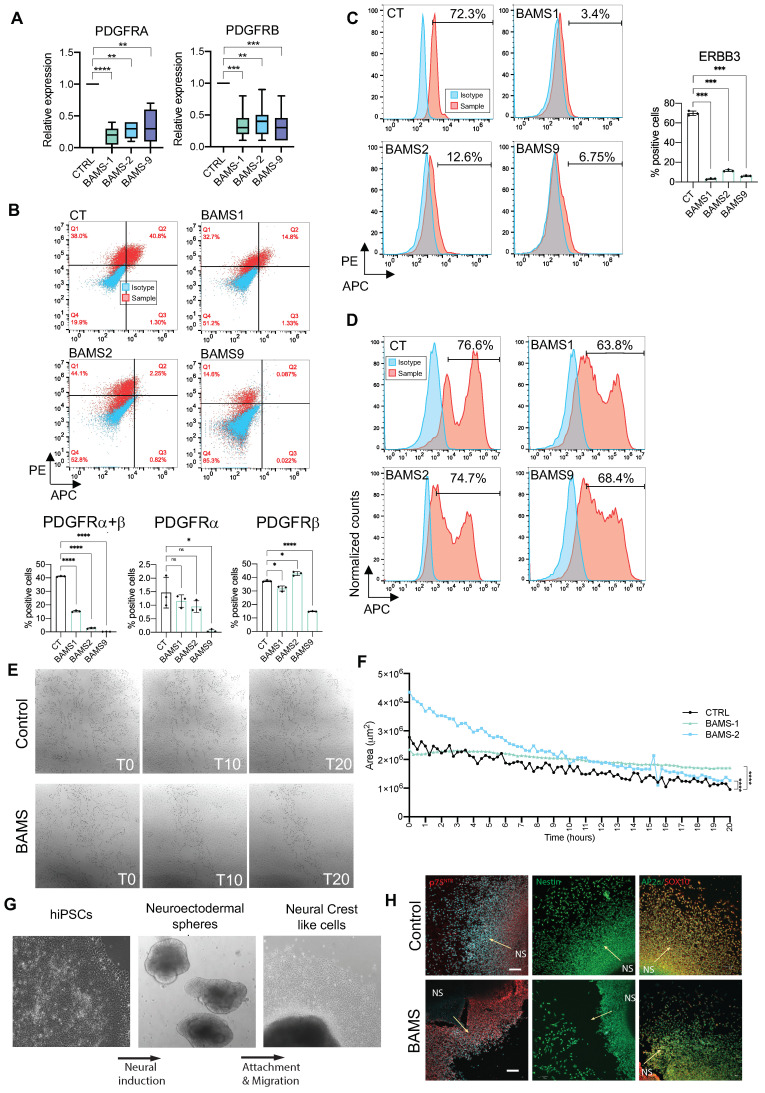
Functional analysis highlights delayed migration in BAMS patients’ cells. (**A**) Expression of *PDGFRα* (Platelet Derived Growth Factor Receptor Alpha) and *β* (Platelet Derived Growth Factor Receptor Alpha Beta) by RT-qPCR in BAMS and control NCSCs. Statistical significance was determined using a Kruskal–Wallis statistical test. ** *p*-value < 0.005, *** *p*-value < 0.0005, and **** *p*-value < 0.00005; (**B**–**D**) Quantification by flow cytometry performed in triplicate for each sample. Statistical significance was determined using Brown–Forsythe and Welch ANOVA tests: * *p*-value < 0.05, **** *p*-value < 0.00005, ns: non-significant. (**B**) Quantification of PDGFRα- and PDGFRβ-positive cells by flow cytometry analysis. APC fluorescence corresponding to PDGFRα is plotted on the *x*-axis and PE fluorescence corresponding to PDGFRβ on the *y*-axis. Fluorescence corresponding to the isotypes, blue dots; samples, red dots. Histograms display the average values and standard deviations for the different biological and technical replicates; (**C**) Quantification of ERBB3-positive NCSCs by flow cytometry analysis. APC fluorescence corresponding to the isotype (blue curve) or ERBB3 antibody (red curve) is plotted on the *x*-axis and normalized counts on the *y*-axis. Histograms display the average values and standard deviations for the different biological and technical replicates; (**D**) Quantification of Ki67-positive NCSCs by flow cytometry analysis. APC fluorescence corresponding to the isotype (blue curve) or Ki67 antibody (red curve) is plotted on the *x*-axis and normalized counts on the *y*-axis. The percentage of Ki67-positve cells is indicated for each sample; (**E**) Representative images of wound closure by migration tested using a scratch assay to determine the rate of migration of NCSCs derived from control (upper panel) or BAMS (lower panel) hiPSCs. Experiments were performed in triplicate from two different differentiation experiments and recording was done by live imaging for 25 h. Several areas (delimited automatically, black lines) were simultaneously analyzed per well. Snapshots correspond to the time at which the scratch was made (T0), after 10 h (T10) and 20 h (T20); (**F**) Quantification of wound closure from the initial scratch in the different conditions. Gap areas were measured in square micrometers (*y*-axis) as a function of time (*x*-axis). Statistical significance was determined using a Friedman non-parametric test for comparison of paired area values between conditions. ****, *p*-value < 0.00005; (**G**) Representative images of human iPSC differentiation into neural crest-like cells through neuroectodermal spheres using protocol adapted from [16]; (**H**) BAMS neural crest generated through neuroectodermal spheres show migration defects. Immunocytochemistry of neural crest markers in control and BAMS neural crest-like cells. BAMS cells show numerous gaps between neuroepithelial spheres (marked by NS) and the emerging neural crest-like cells. Arrows show the direction of neural crest migration from neuroepithelial spheres. Scale bar = 100 μm.

**Table 1 biomedicines-09-00751-t001:** List of hiPSCS clones. All cells are described in [9].

	Diagnosis	Genotype	*SMCHD1* Status	Age	Gender
AG08498	Healthy	>10	No Mutation	1	Male
12759	FSHD1	7RU	No mutation	51	Female
17706	FSHD1	2RU	No mutation	56	Female
	Mosaic	25%			
14586	FSHD2	>10	c.573A > C°; p.Q193P	67	Male
11440	FSHD2	>10	c.2338 + 4A > G; p.S754 *	37	Male
BAMS-1	BAMS	N/A	c.407A > G	5	Male
			p.E136G		
BAMS-2	BAMS	N/A	c.403A > T	28	Female
			p.S135C		
BAMS-9	BAMS	N/A	c.1259A > T	3	Male

*: stop codon.

## Data Availability

The raw RNA-Seq data and raw count matrix were deposited at the NCBI Gene Expression Omnibus (https://www-ncbi-nlm-nih-gov.gate2.inist.fr/geo/ (accessed on 23 June 2021)) under the accession GSE173251.

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
