# Peer review of "AKT Signaling Modifies the Balance between Cell Proliferation and Migration in Neural Crest Cells from Patients Affected with Bosma Arhinia and Microphthalmia Syndrome"

_biomedicines, 2021, doi:10.3390/biomedicines9070751_

Round 1

Reviewer 1 Report

The manuscripts by Laberthonniere et al., shows AKT signaling contributes to neural crest cells migration in patient affected with Bosama Arhinia and Micropthtalmia syndrome.  The authors analyzed FSHD1, 2 and BAMS bulk-RNA sequencing data in depth and identified genes associated with diseases specific and common changes. The used MOCAMUN to find signaling path altered in patient’s cells.  The shows AKT signaling control cell proliferations and migration defect in BAMS.  Authors should address and clarify and following points to improve the article.

Major points:

  1. AKT signaling affected in BAMS patient’s cells. Author should show phosphorylation and total AKT, Pdgfa/b induced AKT signaling in control and BAMS patient’s cells by western blot or FACS.

2. A large part of the computational part of the Paper is based on analysis done with MOGAMUN a, as of yet, not peer-reviewed method. It is out of the scope of our review here to also assess the validity of this tool and so pulls into question how reliable these results are. Overall however, the results seem congruent and supported with orthogonal approaches and consistent within the general story and the approach laid out in the MOGAMUN preprint seems reasonable. It just is not ideal to so heavily rely on such an un-reviewed method.

3. The Pipeline of using STAR and then StringTie (which is often used for Transcfiptome assembly) to merely count Gene counts is a fairly uncommon choice. Using the "--quantMode GeneCounts" functionality of STAR, or using a pseudo-mapper like Salmon and kallisto would be the more appropriate choice. While this choice of pipeline is unlikely to drastically change the results in the present study, the authors should explain their choice for this pipeline and ideally show that it compares favourably to these more common ways of performing the RNA-Seq postprocessing. This might also be considered a minor instead of a major remark.

4. The tool g:Profiler is not mentioned in the methods part (only twice in the results part) and its function is not properly explained.

5. Figure 3C-H and others: The Plot shows edges (connections) between nodes colered in different colors, but the Legend provides no explanation to what is coded with these colors. It provides p-values, but what do these mean? What does each of the shown modules correspond to? Which of these genes are involved in which pathways? What do the colors of the nodes mean? The original method paper explains it, but this paper does not properly! The authors propose interpretations in the Figure legend, that can not be easily confirmed by looking at the Figure itself. i.e. "a large majority of nodes in BAMS and FSHD2 cells highlight changes in transmembrane receptor protein Tyrosine kinase signaling" but the Figure does not show this whatsoever, unless the reader is very familiar with the genes shown and their functions.

Minor:

1. In the methods part section "RNA-Seq data processing and differential expression analysis", the authors mention DESeq2 as being used for DEG Analysis, but later mention edgeR (another tool for DEG Analysis). Which one is it?

2. The FDR cut-off of 0.5 is widespread, but also often criticised (if necessary I can provide citations for this claim) as weak and somewhat arbitrary. How would the results differ, if for example a FDR cut-off of 0.01 were chosen?

3. I would like to encourage the authors to not just mention, that RNA-Seq read QC was performed, but also to provide some important numbers such as adapter contamination and fraction of PCR duplicates.

4. Figure 1D & 2B: Venn Diagramm with this many categories are suboptimal! I recommend the use of an Upset Plot instead!

5. Figure 1E-G: The description of this plot type is repeated multiple times in the Figure Legend. This is not necessary. One explanation would suffice, followed by what specifically is shown in each Plot.

6. Section 3.5 mostly focusses on individual genes, why was the MOGAMUN Analysis relevant then? GO and manual curation would have resulted in the same insight?

7. I assume they provide a complete table of the DEG they find in the Supplements?

8. The explanation of MOGAMUN in the abstract/introduction should be improved. The explanation in the results part is much more straightforward to understand!

And some points out of curiosity:

1. Ectopic Dux4 expression is mentioned (although very low), I would be interested in whether also expression of retrotransposons such as ERV elements, that are known to be regulated by, among others Dux4, also show ectopic expression.

2. I was wondering, whether there is not an established cell line of Neural Crest Cells, that could be manipulated via targeted mutations to introduce the same mutations as in patients.

Author Response

REVIEWER 1

The manuscripts by Laberthonniere et al., shows AKT signaling contributes to neural crest cells migration in patient affected with Bosama Arhinia and Micropthtalmia syndrome.  The authors analyzed FSHD1, 2 and BAMS bulk-RNA sequencing data in depth and identified genes associated with diseases specific and common changes. The used MOCAMUN to find signaling path altered in patient’s cells.  The shows AKT signaling control cell proliferations and migration defect in BAMS.  Authors should address and clarify and following points to improve the article.

Authors’ response: we first thank reviewer 1 for the careful evaluation of our manuscript and constructive suggestions. All comments have been taken into account in the revised version of our manuscript. Our point-by-point response to improve the article is provided below.

Major points:

  1. AKT signaling affected in BAMS patient’s cells. Author should show phosphorylation and total AKT, Pdgfa/b induced AKT signaling in control and BAMS patient’s cells by western blot or FACS.

Authors’ response: We thank reviewer 1 for this interesting comment. Quantification of PDGFRA, PDGFRB and PDGRFRA+B and ERBB3-positive cells are presented in figure 5. We did not succeed in monitoring AKT phosphorylation as activation of the AKT pathways involves many different post-translational modifications (phosphorylation, oxidation, see Risso et al. Biochem J (2015) for review). We did not observe any massive difference among the numerous bands (isoforms) visible after immunoblotting.

In the meantime, we have obtained additional data using a different neural crest cells differentiation protocol that nicely corroborate our initial findings. These data are now presented in figure 5 and described in the results section:We further corroborated these findings using an alternative differentiation protocol for neural crest stem cells based on a neurosphere differentiation step (Figure 5G) [16] by showing a decrease in the overall migration of BAMS-derived cells (Figure 5H).”.

Legend of the corresponding figure (Figure 5):G. Representative images of human iPSC differentiation into neural crest-like cells through neuroectodermal spheres using protocol adapted from [16]. H. BAMS neural crest generated through neuroectodermal spheres show migration defects. Immunocytochemistry of neural crest markers in control and BAMS neural crest-like cells. BAMS cells show numerous gaps between neuroepithelial spheres (marked by NS) and the emerging neural crest-like cells. Arrows show the direction of neural crest migration from neuroepithelial spheres. Scale bar = 100 mm.

The corresponding protocol for differentiation is described in the Materials and Methods section. 

  1. A large part of the computational part of the Paper is based on analysis done with MOGAMUN a, as of yet, not peer-reviewed method. It is out of the scope of our review here to also assess the validity of this tool and so pulls into question how reliable these results are. Overall however, the results seem congruent and supported with orthogonal approaches and consistent within the general story and the approach laid out in the MOGAMUN preprint seems reasonable. It just is not ideal to so heavily rely on such an un-reviewed method. 

Authors’ response: We thank reviewer 1 for raising this point. The manuscript describing and evaluating the MOGAMUN algorithm has been accepted since the submission of this manuscript (acceptation letter 25/05/2021) and will be available in PLOS Computational Biology very soon. In addition, the MOGAMUN code is freely available from GitHub for reproducibility, and the MOGAMUN package has been accepted in Bioconductor.

Additional information regarding the advantages of MOGAMUM have been added in the introduction: “We further analyzed differentially expressed genes with MOGAMUN, an algorithm that uses a multi-objective genetic algorithm to extract active modules by integrating expression data with multiple biological networks. MOGAMUN optimizes two objective functions, one related to interaction density and one related to gene expression deregulation to reveal active modules, i.e, subnetworks differentially regulated between conditions.”

  1. The Pipeline of using STAR and then StringTie (which is often used for Transcfiptome assembly) to merely count Gene counts is a fairly uncommon choice. Using the "--quantMode GeneCounts" functionality of STAR, or using a pseudo-mapper like Salmon and kallisto would be the more appropriate choice. While this choice of pipeline is unlikely to drastically change the results in the present study, the authors should explain their choice for this pipeline and ideally show that it compares favourably to these more common ways of performing the RNA-Seq postprocessing. This might also be considered a minor instead of a major remark. 

Authors’ response: The pipeline used in our manuscript is based on the selection of genes differentially expressed between conditions, determined using DeSeq2. While it is possible to quantify gene counts using -quantMode GeneCounts from STAR, such an approach would output normalized counts such as TPM.

Even though Stringtie main design lies in transcriptome assembly, its features allows this kind of analysis and is suitable for downstream analysis on mainstream differential expression tools (Pertea et al. Nature Protocols, 2016).

Concerning pseudo-mappers like Salmon and Kallisto, they would not be pertinent after having already performed the alignment as one of their main benefits are computing time when compared to classical mapping tools. As mentioned by the reviewer, the choice of the analysis pipeline is not the most critical step in the RNA-seq processing and may not have major influence on the results of the analysis.

  1. The tool g:Profiler is not mentioned in the methods part (only twice in the results part) and its function is not properly explained. 

Authors’ response: A proper description of g:Profiler has been added to the methods part. This description indicates “Genes corresponding to each nodes were analyzed using g:Profiler (https://biit.cs.ut.ee/gprofiler) to define the corresponding molecular function and corresponding p-value. g:Profiler takes as input a list of genes, uses their annotation (from Ensembl database), and performs an hypergeometric test to find over-representation (i.e., enrichment) of functional terms (e.g., biological processes, molecular functions, cellular components, etc.). Here we used the g:GOSt function to perform the functional enrichment analysis of individual gene lists.”.

  1. Figure 3C-H and others: The Plot shows edges (connections) between nodes colered in different colors, but the Legend provides no explanation to what is coded with these colors. It provides p-values, but what do these mean? What does each of the shown modules correspond to? Which of these genes are involved in which pathways? What do the colors of the nodes mean? The original method paper explains it, but this paper does not properly! The authors propose interpretations in the Figure legend, that can not be easily confirmed by looking at the Figure itself. i.e. "a large majority of nodes in BAMS and FSHD2 cells highlight changes in transmembrane receptor protein Tyrosine kinase signaling" but the Figure does not show this whatsoever, unless the reader is very familiar with the genes shown and their functions. 

The color codes of the edges correspond to the interaction sources. This information is provided in the figure legends: “The color of the edges (links) denotes the type of interaction/relationship between each pair of genes. Specifically, protein-protein interactions (blue links), biological pathways (orange links) and correlation of expression data (yellow links). Up-regulated nodes are colored in red and down-regulated ones, in green, the intensity of the color reflects the fold-change. Nodes with a dark border are significant (i.e., FDR<0.05 and -2>FC>2). »

Authors’ response: We apologize for the lack of detail regarding the information provided regarding the active modules retrieved by MOGAMUN. Information on the color codes is given in the legend of the corresponding figures and has also been added in the result section (paragraph 3.4): “For our experiments, we considered protein-protein interactions (blue edges), biological pathways (red edges) and correlation of expression (yellow edges) in the multiplex biological network.”.

The set of genes belonging to each active modules were used as input for g:Profiler to identify significantly enriched Biological pathways (BP) and processes. p-values correspond to the level of significance of the most significant BP identified using g:Profiler for the genes composing the individual active modules. This information is provided in the legend of figures 3 and 4: “The list of genes in each active module was analyzed using g:Profiler to define the corresponding molecular function and p-value ».

The most significant biological process corresponding to each module presented in each figure is indicated in the legend of the figure.

The meaning of the colors is indicated as well.

We agree that the interpretation of the figures might be complex for readers who are not familiar with genes present in the active modules and their function. Regarding Tyrosine Kinase signaling, the list of related genes is provided in the legend of figures 4.

Minor: 

1. In the methods part section "RNA-Seq data processing and differential expression analysis", the authors mention DESeq2 as being used for DEG Analysis, but later mention edgeR (another tool for DEG Analysis). Which one is it?

Authors’ response: We apologize for this mistake. DESeq2 was used all along. This has been corrected in the materials and methods section in the revised manuscript.

  1. The FDR cut-off of 0.5 is widespread, but also often criticised (if necessary I can provide citations for this claim) as weak and somewhat arbitrary. How would the results differ, if for example a FDR cut-off of 0.01 were chosen?

Authors’ response: We agree with the reviewer that this threshold is the most frequently used but without clear bases. We checked the number of DEGs that are obtained using a threshold of FDR of 0.01.

Number of DEG at 0.05 FDR threshold plus abs(log2FC) > 2

4220 DEGs in BAMS_vs_controls

3775 DEGS in FSHD1 vs controls

3500 DEGS in FSHD1 mosaic vs its isogenic control

3610 DEGS in FSHD2 vs controls

Number of DEG at 0.01 FDR threshold plus abs(log2FC) > 2

3585 DEGs in BAMS_vs_controls

3131 DEGS in FSHD1 vs controls

2799 DEGS in FSHD1 mosaic vs its isogenic control

3025 DEGS in FSHD2 vs controls

Changing the 0.05 threshold for a more stringent threshold of 0.01 does not drastically change the number of obtained DEGs; DEGs retrieved using a 0.05 threshold represent 80-85% of the DEGs retrieved using a 0.01 threshold, as presented above.

In enrichment analyses, using the most stringent DEGs dataset would modulate the ratio of entry genes on background universe and the computation of the statistical enrichment tests, but since the p-value is strong we do not expect any drastic changes in the output.

  1. I would like to encourage the authors to not just mention, that RNA-Seq read QC was performed, but also to provide some important numbers such as adapter contamination and fraction of PCR duplicates. 

Authors’ response: We did not find any sample with an adapter contamination > 0.1%. All samples had less than 1% of reads corresponding to overrepresented sequences. The % of bases above Q30 > 96.87 for all samples. Duplicated sequences were marked using Sambamba (v0.6.6) and excluded from final counting.

This information was added in the legend of supplementary figure 2

  1. Figure 1D & 2B: Venn Diagramm with this many categories are suboptimal! I recommend the use of an Upset Plot instead!

Authors’ response: Venn diagram are the most commonly used representation for genes list crossing, we admit that it might be difficult to read. That’s why numbers that were discussed in the text were outlined using a black circle.

However, as recommended by Reviewer 1, representations have been modified throughout the manuscript and genes list crossing are now represented with Upset Plots. The legends of the corresponding figures (figures 1 and 2) have been modified accordingly.

  1. Figure 1E-G: The description of this plot type is repeated multiple times in the Figure Legend. This is not necessary. One explanation would suffice, followed by what specifically is shown in each plot.

Authors’ response: The legend of the figure has been modified accordingly

  1. Section 3.5 mostly focusses on individual genes, why was the MOGAMUN Analysis relevant then? GO and manual curation would have resulted in the same insight? 

Authors’ response: The main goal of MOGAMUN is to help identifying genes of interest. We went from a list of thousands of significantly Differentially Expressed Genes (DEG) to 88 genes of interest (out of which, 56 are DEGs) in the case of BAMS, and 111 genes of interest (67 DEGs) for FSHD2. GO enrichment would have focused only on annotated genes, and manual curation would have been tedious. The MOGAMUN approach considering different sources of physical and functional interactions allows integrating previous knowledge and helps the users to focalize on relevant subparts of the differential expression dataset.

  1. I assume they provide a complete table of the DEG they find in the Supplements? 

Authors’ response: Indeed, a complete table of DEGs table is available in the supplemental information.

  1. The explanation of MOGAMUN in the abstract/introduction should be improved. The explanation in the results part is much more straightforward to understand! 

Authors’ response: Changes have been made in the abstract and in the introduction to better explain the principles of MOGAMUN; “

In the abstract: “Besides classical differential expression analyses, we analyzed our data using MOGAMUN, an algorithm allowing the extraction of active module by integrating differential expression data with biological networks”.

And in the introduction: “We further analyzed differentially expressed genes with MOGAMUN, an algorithm that uses a multi-objective genetic algorithm to extract active modules by integrating expression data with multiple biological networks. MOGAMUN optimizes two objective functions, one related to interaction density and one related to gene expression deregulation. It reveals active modules, i.e, subnetworks differentially regulated between conditions.”

And some points out of curiosity: 

  1. Ectopic Dux4 expression is mentioned (although very low), I would be interested in whether also expression of retrotransposons such as ERV elements, that are known to be regulated by, among others Dux4, also show ectopic expression. 

Authors’ response: This is indeed a very interesting question. The RNA Seq was performed on polyA+ transcript and we did not observe any significant enrichment in ERV transcripts in the list of genes. We agree that this would be very interesting to monitor.

  1. I was wondering, whether there is not an established cell line of Neural Crest Cells that could be manipulated via targeted mutations to introduce the same mutations as in patients. 

Authors’ response: We agree that indeed this would be interesting to perform this type of experiments but instead of using “engineered cellular models” we decided to restrict our analysis to patient’s cells. Manipulated Neural Crest Cells will be likely very informative in the near future for more in depth mechanistic investigations.

Reviewer 2 Report

The authors demonstrated that the relationship of neural crest stem cell (NSCS) migration and proliferation in the type 2 Facio Scapulo Humeral Dystrophy (FSHD2) and Bosma Arhinia and Microphtalmia syndrome (BAMS) by using human induced pluripotent stem cell-derived NSCSs. Through comparison of RNA-seq data, the authors selected genes with differences and changes. This is a very meaningful study on the issue of rare disease. Although this study contains an impressive information of FSHD2-BAMS, the structure and methodological procedure is not coherent and not clear enough to allow a detailed assessment of the study. As it is, many aspects within the study need addressing to be considered for publication in BIOMEDICINE. Because there is no uploading files for supplementary figures, we cannot evaluate this study.

  1. Abstract: adding SMCHD1 full name
  2. In abstract, the author demonstrated that AKT signaling controls the cell proliferation-migration balance, however title was not represented. Title should be changed in this context.
  3. Page 3, lane 3: Using superscript, cm2. Adding hESC, DMEM, KSR, NEAA and FGF full name
  4. In neural crest stem cell differentiation, what is the mixture ratio of N2 and KSR medium up to D10?
  5. Is MOGAMUN a program devised by the authors? Or is it a commercially available program? Also, has MOGAMUN been verified by an accredited agency?
  6. Section of wound healing closure scratch test: adding NCSC and MRI full name. Using superscript, cm2.
  7. In this study, what kind of statistical analysis did the author use? The author should create a section in the method and describe it.
  8. Section of flow cytometry analysis: adding SVF, APC and PDGFR full name
  9. In experimental design, did the author consider “age” to interpret the results? As the author can see, it consists of a wide range of ages, from fetal to old age.
  10. In Figure 1A, it appears that RNA-seq has been analyzed serially (control, fibroblast, hPSC and NCSCs).
  11. There is no explanation of immonostain method implemented in Figure 1C.

Author Response

REVIEWER 2

The authors demonstrated that the relationship of neural crest stem cell (NSCS) migration and proliferation in the type 2 Facio Scapulo Humeral Dystrophy (FSHD2) and Bosma Arhinia and Microphtalmia syndrome (BAMS) by using human induced pluripotent stem cell-derived NSCSs. Through comparison of RNA-seq data, the authors selected genes with differences and changes. This is a very meaningful study on the issue of rare disease. Although this study contains an impressive information of FSHD2-BAMS, the structure and methodological procedure is not coherent and not clear enough to allow a detailed assessment of the study. As it is, many aspects within the study need addressing to be considered for publication in BIOMEDICINE. Because there is no uploading files for supplementary figures, we cannot evaluate this study.

Authors’ response: We first thank reviewer 2 for the careful evaluation of our manuscript and positive comments on the importance of the work and questions raised. We apologize for the issue with access to Supplemental information that were downloaded on the journal’s website at the time of submission. All comments have been taken into account in the revised version of our manuscript. Our point-by-point response to improve the article is provided below.

  1. Abstract: adding SMCHD1 full name

Authors’ response: This has been modified line 20-21, as requested.

  1. In abstract, the author demonstrated that AKT signaling controls the cell proliferation-migration balance, however title was not represented. Title should be changed in this context.

Authors’ response: We agree with this comment and the title has been modified accoridingly: “AKT signaling modifies the balance between cell proliferation and migration in neural crest cells from patients affected with Bosma Arhinia and Microphtalmia syndrome

  1. Page 3, lane 3: Using superscript, cm2. Adding hESC, DMEM, KSR, NEAA and FGF full name

Authors’ response: Full names have been added in the Materials and Methods section, lines 108 to 111 and Superscript used as recommanded.

  1. In neural crest stem cell differentiation, what is the mixture ratio of N2 and KSR medium up to D10?

Authors’ response: We apologize for this lack of details. The materials and Methods section has been modified to “The KSR medium is gradually replaced by the N2 medium. Each day, the culture medium is changed with a KSR/N2 mixture with the proportion of KSR decreasing and that of N2 increasing to reach 100% N2 at D10. The corresponding sentence has been modified in the Materials and Methods section: “From D4 to D11, N2 (DMEM/F12, 0,15% Glucose, 1% N2, 20µg/mL insulin, 5mM HEPES) is progressively mixed (starting with 15% of N2 at D4) with the KSR medium in order to reach 100% of N2 medium at D10. Cells are collected at D11.”.

  1. Is MOGAMUN a program devised by the authors? Or is it a commercially available program? Also, has MOGAMUN been verified by an accredited agency?

Authors’ response: MOGAMUN is an open source Bioconductor package (https://bioconductor.org/packages/release/bioc/html/MOGAMUN.html) designed and implemented by two of the authors (EMNT and AB). The code is fully available from GitHub for reproducibility. The associated paper has been accepted in PLOS Computational Biology since the submission of this manuscript and will be available soon.

  1. Section of wound healing closure scratch test: adding NCSC and MRI full name. Using superscript, cm2.

Authors’ response: Full names were added in the text and cm2 indicated with a supersecript

  1. In this study, what kind of statistical analysis did the author use? The author should create a section in the method and describe it.

Authors’ response: Statistical analysis performed are described in methods and in the legend of each figure.

  1. Section of flow cytometry analysis: adding SVF, APC and PDGFR full name

Authors’ response: Full names were added to the text

  1. In experimental design, did the author consider “age” to interpret the results? As the author can see, it consists of a wide range of ages, from fetal to old age.

  1. In Figure 1A, it appears that RNA-seq has been analyzed serially (control, fibroblast, hPSC and NCSCs).

Authors’ response: In this study, RNAseq was performed on NCSCs-derive hiPCs, Figure1A was confusing and has been changed.

  1. There is no explanation of immonostain method implemented in Figure 1C.

Authors’ response: We apologize for this omission in the materials and Methods section. Immunostaining method was added in the materials and methods section.

Round 2

Reviewer 2 Report

This paper has been completely revised as suggested by the reviewer. I wish this study may have a great suggestion to their area.